# Nutritional Treatment as a Synergic Intervention to Pharmacological Therapy in CKD Patients

**DOI:** 10.3390/nu15122715

**Published:** 2023-06-12

**Authors:** Domenico Giannese, Claudia D’Alessandro, Vincenzo Panichi, Nicola Pellegrino, Adamasco Cupisti

**Affiliations:** Department of Clinical and Experimental Medicine, University of Pisa, 56126 Pisa, Italy; domenico.giannese@phd.unipi.it (D.G.); claudia.dalessandro@unipi.it (C.D.); vincenzo.panichi@unipi.it (V.P.); n.pellegrino1@studenti.unipi.it (N.P.)

**Keywords:** low protein diet, low sodium diet, nutrition, CKD, proteinuria, SGLT2i, phosphate

## Abstract

Nutritional and pharmacological therapies represent the basis for non-dialysis management of CKD patients. Both kinds of treatments have specific and unchangeable features and, in certain cases, they also have a synergic action. For instance, dietary sodium restriction enhances the anti-proteinuric and anti-hypertensive effects of RAAS inhibitors, low protein intake reduces insulin resistance and enhances responsiveness to epoetin therapy, and phosphate restriction cooperates with phosphate binders to reduce the net phosphate intake and its consequences on mineral metabolism. It can also be speculated that a reduction in either protein or salt intake can potentially amplify the anti-proteinuric and reno-protective effects of SGLT2 inhibitors. Therefore, the synergic use of nutritional therapy and medications optimizes CKD treatment. Quality of care management is improved and becomes more effective when compared to either treatment alone, with lower costs and fewer risks of unwanted side effects. This narrative review summarizes the established evidence of the synergistic action carried out by the combination of nutritional and pharmacological treatments, underlying how they are not alternative but complementary in CKD patient care.

## 1. Introduction

Chronic kidney disease (CKD) is quite a prevalent disease, affecting up to 10% of the general population, and it is associated with a high rate of morbidity and mortality. By 2040, CKD is predicted to be the fifth leading cause of years of life lost globally [1]. It is thus evident that CKD represents a real concern for public health and for the economy.

Patients with CKD have an increased risk of cardiovascular diseases and events and progression towards end stage kidney disease (ESKD). Hypertension, proteinuria, anemia and CKD-associated disorders of the mineral and bone metabolism (CKD-MBD) represent major factors influencing cardiovascular and kidney damage progression. All of these modifiable factors are addressed by means of pharmacological therapy. Namely, renin angiotensin aldosterone system (RAAS) inhibitors, sodium glucose transporter 2 (SGLT2) inhibitors, erythropoiesis stimulating agents (ESAs), intestinal phosphate binders and vitamin D derivatives are currently included in the pharmacological treatment of CKD patients.

From the 1960s onwards, dietary protein restriction was the sole approach used to correct uremia-related signs and symptoms and, in this way, to delay the need for dialysis. Instead, protein restriction seemed to have a statistically significant but small and clinically negligible effect on slowing the rate of decline in glomerular filtration rate (GFR) [2]. The mainstay of dietary treatment was the modulation of protein intake up to severe restriction and the reduction in dietary sodium and phosphate load, while maintaining an adequate energy supply [3]. In the 1990s, the availability of new medications in clinical practice seemed more attractive for nephrologists and patients than dietary interventions, because of obvious concerns about adherence to dietary restrictions. Despite this, dietary interventions currently remain the only unchangeable tool to reduce uremic intoxications and maintain nutritional status [4], while strong evidence exists that drugs can reduce the progression of the disease more effectively than protein restriction [5,6]. However, nutritional interventions can also be important because they make several drugs used in CKD patients more effective. This review discusses the role of dietary treatment in facilitating the effects of the pharmacological therapy administered to CKD patients.

## 2. RAAS Blockers

Proteinuria is the major risk factor in the progression of kidney damage [7], as well acting as a marker of severity of the disease and a target for therapy. In the presence of increased glomerular blood pressure and permeability to albumin, the increased amount of albumin in the ultrafiltrate is reabsorbed by tubular cells and digested in the lysosomes. This process is deleterious for both glomeruli and tubulo-interstitium, finally leading to glomerulosclerosis and tubule-interstitial fibrosis [8]. In subtotally nephrectomized rats, compensative adaptation of the remaining glomeruli occurs owing to the increase in capillary hydraulic pressure that favors protein passage through the glomerular capillary wall [9], resulting in progressive glomerulosclerosis [10,11]. Moreover, the overload of proteins in the tubular lumen—in particular, albumin, IgG, and transferrin—promotes the release of profibrotic and pro-inflammatory molecules, inducing the up-regulation or down-regulation of more than 160 genes in kidney tubule cells [12]. The final mechanism results in an abnormal accumulation collagen, fibronectin and other proteins that leads to interstitial fibrosis [13]. Tubulo-interstitial damage is also due to the peptides produced by dendritic cells for the degradation of albumin [14], and it is also accelerated by direct damage to the tubuli mediated by complementary factors, growth factors and pro-inflammatory molecules [15]. So, the lower the proteinuria is, the lower the chance of progressive kidney damage is.

Angiotensin-converting enzyme (ACE) inhibitors and angiotensin-receptor blockers (ARB) are well-known anti-hypertensive and urine protein-lowering agents. Owing to these features, they are a major reno-protective class of medications. Apart from their anti-hypertensive action and anti-inflammatory, antioxidant and anti-proliferative properties, the effects of RAAS inhibitors include changes in glomerular hemodynamics. Namely, they reduce the vasomotor tone of the efferent arteriole, which reduces intraglomerular pressure and increases renal plasma flow, with a consequent reduction in the filtration fraction. A direct consequence is the reduction in the trans glomerular traffic of proteins and of sheer stress, which, in turn, can activate the local production of growth factors, pro-inflammatory and profibrotic cytokines. This is the reason why RAAS blockers are considered as the first-line therapy for treating hypertension and proteinuria in CKD patients, therefore acting as major renoprotective agents.

It is noteworthy that the efficacy of RAAS inhibitors is affected by dietary factors. The anti-hypertensive and anti-proteinuric efficacy of ACE inhibitors in CKD patients is blunted by high dietary sodium intake. A post hoc analysis of the first and second Ramipril Efficacy in Nephropathy (REIN) [16] studies examined the progression to ESKD among 500 CKD non-diabetic patients treated with 5 mg Ramipril based on the dietary sodium intake that was assumed to be equal to the 24 h renal excretion. Urinary sodium was normalized by urine creatinine excretion and defined as low (<100 mEq/g), medium (100–200 mEq/g) or high (≥200 mEq/g). During an average follow-up of more than 4 years, the incidence of ESKD was 6.1, 7.9 and 18.2 per 100 patient-years in the low, medium, and high sodium intake groups, respectively. It was estimated that for each 100 mEq sodium/g creatinine urinary excretion, a 1.61-times higher risk for ESKD occurred. This association was independent of arterial blood pressure values. The authors concluded that among non-diabetic CKD patients, high dietary salt intake, over 14 g daily, blunted the anti-proteinuric effect of ACE inhibitor treatment and increased the risk for ESKD, independently of arterial blood pressure control [17]. A post hoc analysis of the RENAAL [18] and IDNT [19] trials was performed to assess the effect of sodium intake in 1177 type 2 diabetic patients with CKD who were randomized to receive ARB therapy or non-RAAS blocker-based antihypertensive therapy. Renal and cardiovascular outcomes were compared as a function of dietary sodium intake during treatment, which was estimated according to the 24 h urinary sodium/creatinine ratio. The greatest favorable long-term effects on renal and cardiovascular events occurred in the patients undergoing ARB treatment, when compared to those on non-RAAS blocker-based therapy and only on low sodium intake. Compared to non-RAAS blockers, the trend of the risk for renal events was significantly reduced by 43%, not changed, or increased by 37% for each tertile of sodium intake, respectively. Similarly, the risk for cardiovascular events was significantly reduced by 37% only in the lowest tertile of sodium intake. For both renal and cardiovascular events, the risk was increased in the ARB-treated group at the highest tertile of sodium intake. As a whole, the effects of ARB on renal and cardiovascular outcomes were greater in patients with type 2 diabetic nephropathy when dietary sodium intake was the lowest. These two post hoc analysis studies underscore the need to avoid high sodium intake in ARB-treated non-diabetic and type 2 diabetic CKD patients [20]. A low-sodium diet exerts anti-proteinuric and anti-hypertensive effects, but lowering proteinuria by means of sodium restriction was independent of the changes in arterial blood pressure [17]

A multicenter cross-over randomized controlled trial including 52 patients with non-diabetic nephropathy on ACE inhibition therapy compared the effects of the addition of dietary sodium restriction or angiotensin receptor blockade on urine protein excretion and on arterial blood pressure. Sodium urine excretion was assumed as a measure of dietary sodium intake: a regular sodium diet was targeted at 200 mmol sodium daily, whereas a low-sodium diet was targeted at 50 mmol sodium daily. All patients were treated for four 6-week periods, in random order, with ARB (i.e., valsartan 320 mg/day) on top of ACE-inhibitor treatment (lisinopril 40 mg/day). Drug interventions were conducted on a double-blind basis; dietary interventions were open label. The addition of a low-sodium diet to ACE inhibition resulted in a 51% reduction in proteinuria, which was significantly greater than the reduction in proteinuria provided by the addition of ARB (21%) to a regular sodium diet. Mean systolic blood pressure was not significantly reduced by the addition of ARB, but it was lowered by the addition of a low-sodium diet and ARB plus a low sodium diet on top of ACE inhibition. The reduction in systolic blood pressure caused by the addition of a low-sodium diet was significantly greater than that occurring due to the addition of ARB and was similar to the reduction in systolic blood pressure caused by the addition of ARB and a low-sodium diet to ACE inhibition [21]. The results of this thorough study demonstrated that dietary sodium restriction was more effective than the dual blockade of RAAS for lowering urine protein excretion and systolic blood pressure in CKD patients, supporting the relevant role of sodium restriction in the management of CKD patients. Moreover, dual ACEi/ARB blockade is highly questionable and no longer recommended in CKD patients especially among diabetics, mainly due to the increased risk of hyperkalemia [22].

By interfering with the adaptation mechanisms to changes in body sodium, RAAS blockade makes the arterial blood pressure more sensitive to sodium status. In this way, sodium restriction enhances the response to RAAS blockade on both blood pressure and proteinuria [23,24].

Lowering sodium serum levels may occur in patients taking ACEI or ARB [25]. In addition, low salt intake can lead to hyponatremia [26], but when associated with high water intake, it can lead to an impairment of the water and sodium balance and hence of body water homeostasis. The risk of hyponatremia is more consistent when a low-sodium diet is associated with a high dose of diuretics. Monitoring of the sodium serum level is recommended in patients with ACE inhibitor or ARB administration and on a low-sodium diet. We did not find a special warning in the literature about the association between RAAS inhibitors and sodium restriction, which, on the contrary, is strongly recommended.

Since the 1980s, it has been demonstrated that both the quality and quantity of protein intake can modulate glomerular hemodynamics. A high protein intake induces the release of local and humoral mediators causing hyperfiltration [27]. In particular, a high-protein diet stimulates the production of eicosanoids [28] and nitric oxide [29] that cause vasodilatation in the afferent arteriole and increase glomerular blood flow and intraglomerular pressure. On the contrary, a low-protein diet results in a reduction in single nephron GFR and glomerular hypertension. The consequent reduction in hyperfiltration and proteinuria may slow the decline of residual kidney function [9,30]. Kontessis et al. showed that meat intake induced an acute increase in GFR and renal plasma flow, but this response was blunted using the same amount of protein from soy [31]. So, also the quality of dietary protein differently affects glomerular hemodynamics.

The hypothesis thus arose that the glomerular changes induced by RAAS inhibitors and protein intakes might be additive. Ruilope et al. reported that nephrotic patients with preserved renal function showed a decrease in proteinuria when shifted from a protein intake of 1.0 g/kg/d to 0.3 g/kg/d [32]. The administration of an ACE inhibitor on top of the protein-restricted diet resulted in a further decrease in proteinuria. The results demonstrated the additive antiproteinuric effect of a low protein intake associated with ACE inhibition.

In a retrospective cohort study of 78 proteinuric patients with stage 4–5 CKD, proteinuria decreased significantly after patients were treated with a 0.4 g/kg/d protein diet supplemented with essential amino acids and keto-analogues. The reduction was similar in patients receiving and not receiving ACE inhibition therapy. These data support the hypothesis that the anti-proteinuric effect of low-protein diets and ACE inhibitors have different mechanisms and may be additive [33]. It is relatively obvious that very-low-protein diets are difficult to adhere to, and this is the reason why they are only proposed to motivated and selected patients. When diet management is correct, no signs of malnutrition occur [34,35].

## 3. SGLT2i

The recent literature turns the spotlight on new antidiabetic drugs and their effects on CKD patients, i.e., sodium-glucose-transporter 2 (SGLT2) inhibitors. The nephroprotection and antiproteinuric effects, independently from achieving good glucose control, are now well known: in randomized controlled trials, the use of SLGT2 inhibitors reduces progression to kidney disease, reduces hospitalizations and cardiovascular death, particularly in the case of overt proteinuria [36,37]. The favorable effects of SGLT2 inhibitors on the kidney consist of a reduction in glomerular hyperfiltration via blocking sodium-glucose cotransporters and the consequent reduction in sodium reabsorption as well. It follows that the juxtaglomerular apparatus detects the increase in sodium load in the distal tubule and induces afferent arteriolar vasoconstriction, namely the activation of tubulo-glomerular feedback. As a result, renal plasma flow and intraglomerular pressure and filtration are reduced. It is of great interest to hypothesize a combined use of SGLT2 inhibitors and protein restriction, considering the potential synergistic effect on glomerular hemodynamics [38]. In order to understand this promising interaction, van der Aart-van der Beek et al. evaluated, by means of a post hoc analysis, SGLT2 inhibitors’ effects at different levels of protein intake, in three randomized controlled trials [39,40,41] in proteinuric CKD patients, with or without diabetes, receiving dapagliflozin [42]. The results showed that the effect on the initial drop in GFR and on the reduction in the urinary albumin/creatinine ratio are independent of protein intake, namely 58.4, 63.6, and 90.0 g/d as median values. Based on these findings, Kalantar-Zadeh et al. speculated that biologically plausible trends exist suggesting that a low-protein diet may enhance the effect of SGLT2 inhibitors on albuminuria (39 and 38% of extra effect) [43].

In addition, low salt intake mitigated the drop in eGFR that occurs at the start of therapy and improved the long-term positive effect on the eGFR of SGLT2 inhibitors, suggesting a comprehensive nutritional approach to improve renal outcomes [44]. These suggestions in the literature pave the way for new studies on the combined use of SGLT2 inhibitors and a synergic dietary approach on glomerular blood flow and hyperfiltration for reducing proteinuria and slowing CKD progression.

The potential benefits of using a combination of SGLT2 inhibitors and protein and/or salt restriction are not limited to the glomerular hemodynamic effects. In fact, both treatments act on the metabolism of the kidney and on the autophagy process. It has been reported that protein restriction can restore autophagy in diabetic mice, ameliorates tubulointerstitial damage and kidney function [45]. Similarly, SGLT2 inhibitors modulate autophagy by means of a direct effect on autophagosomes and autolysosomes [46].

As is indeed known, autophagy has a critical role in CKD: it allows the maintenance of cellular integrity, function and homeostasis via the degradation of cytoplasmatic components and their reuse, namely through normal cell turnover. In cases of dysregulation, acute kidney injury and incomplete repair after damage are favored. The advanced glycation end-products induced injury of mesangial cell in diabetic nephropathy and autophagy counteracts this. Moreover, podocytes have abundant autophagosomes and, in case of impaired autophagy, proteinuria, glomerulopathy and kidney failure can occur. Last but not least, autophagy is also essential for the integrity of proximal tubules and for limiting tubulointerstitial sclerotic evolution [47].

SGLT2 inhibitor therapy can also contribute to body weight loss and possibly to promoting protein catabolism. In this case, the use of SGLT2 inhibitors alongside protein restriction might be of concern, particularly in elderly patients with CKD.

The effects of SGLT2 inhibition on muscle mass are not yet definite; decreases and no change in skeletal muscle mass have been reported. Negative energy balance increases circulating amino acid concentrations and stimulates amino acid oxidation. These findings may be an expression of protein catabolism, similarly to the increased fat oxidation. Dapagliflozin treatment causes a reduction in body weight mainly by reducing fat mass, but some changes are reported also in amino acid metabolism [48]. The study by Horibe et al. reported that the addition of dapagliflozin significantly reduced body and fat mass in patients with type 2 diabetes not receiving insulin. Instead, skeletal muscle mass showed no significant reduction after treatment with dapagliflozin for 24-weeks. Finally, changes in the plasma amino acid profile were detected after a 24-week treatment with dapagliflozin. A reduction in skeletal muscle mass caused by SGLT2 inhibitors might have a negative effect in older or lean patients with type 2 diabetes. The nutritional care management of CKD patients, especially when they are older, must include the monitoring of nutritional state and body composition.

## 4. Erythropoietin

Anemia is a common complication in patients with CKD, especially in stage IV and V. Its prevalence exceeds 50% of patients [49], and it is associated with adverse outcomes and a low quality of life [50]. The etiology of anemia, which presents itself as normochromic and normocytic, is not only traced to erythropoietin deficiency. A complex inflammatory interaction due to IL1, IL2, TNFα, and INFy [51], the effects of hepcidin on iron metabolism [52], the altered intestinal absorption of nutrients, which is essential in erythrocyte maturation and production, and altered hormonal/receptor responses [53] are all additional factors which contribute to the impaired response of bone marrow to erythropoietin and thus to the onset of anemia in CKD. Since 1989, erythropoietin stimulating agents (ESA) and, most recently, the hypoxia inducible factor prolyl-hydroxylase inhibitors (HIF-PHIs) have deeply changed the management of [54]. Some patients require higher doses of ESA, until a point at which they can be defined as ESA-hyporesponsive according to the KDIGO guidelines [55]. The mechanisms sustaining the ineffective response to therapy are directly or indirectly linked to CKD: inflammation, poor control of CKD-MBD, iron deficiency and malnutrition (folic acid deficiency, vitamin C and B deficiency, alpha-lipoic acid and L-Carnitine deficiency), as well as the retention of uremic toxins that impair the erythropoiesis process [56]. A low-protein diet plays a role in reducing inflammation and hepcidin level and improving malnutrition and the deficiency of nutrients also by interacting with microbiota as well as reducing the production of nitrogen waste products. In fact, the benefits of diets based on protein restriction for CKD patients may be related not only to the level of serum urea, but in general to the all-round treatment of CKD and its consequences. Di Iorio et al. demonstrated the reduction in ESA dosage in patients treated with a very-low-protein diet supplemented with ketoanalogues and essential amino acids [57]. In particular, the authors found an inverse relationship between the good response to ESA and the level of PTH, demonstrating the role of the diet in the synergic management of CKD-related anemia with ESA. It may be speculated that a favorable effect may also occur with HIF-PHIs. The lower the dose of EPO that reaches the target is, the lower the cost of therapy; this element further underlines that dietary protein restriction makes it possible to also save money.

## 5. Intestinal Phosphate Binders

The term CKD-MBD indicates CKD-associated disorders of the mineral and bone metabolism. It includes biochemical changes regarding serum phosphate, calcium, PTH and vitamin D, bone structure remodeling abnormalities and vascular or soft tissue calcifications.

In the pathogenesis of CKD-MBD, a pivotal role is played by fibroblastic growth factor 23 (FGF23). FGF23 is a fosfatonin produced by osteocytes in response to a dietary load of phosphorus, or an increase in phosphorus or calcitriol serum levels. The revised theory of secondary hyperparathyroidism has, as first step, a dietary phosphate load, in the presence of a kidney damage. Under the conditions of GFR loss, the kidney’s capacity to manage a high phosphorus load may be impaired; for this reason, a stimulus to reduce phosphorus tubular reabsorption leading to increased phosphate excretion for a single nephron is needed. This is obtained by means of FGF23 production, together with the co-factor Klotho [58]. FGF23 has been considered as a factor associated with increased morbidity and mortality both in dialysis and in CKD patients, independent of plasma phosphate levels. It is also associated with left ventricular hypertrophy in normal subjects, heart failure and stroke [59]. Moreover, high levels of FGF23 in CKD patients are independent predictors of kidney damage progression [60]. For all of these reasons, the prevention and treatment of CKD must begin from the dietary control of phosphorus intake load. This can be obtained in different ways, namely the absolute content of foods, phosphorus bio-availability, and changes in cooking methods, which are graphically summarized in the phosphorus pyramid [61,62]. To reduce the net intestinal absorption of phosphate, phosphate binders are largely used in CKD and ESKD patients. The currently available intestinal phosphate binders differ among themselves in terms of tolerability, efficacy, cost, and calcium or aluminum content. The newer ones are the most effective, supplying no calcium and no aluminum, but the absolute amount of phosphate that is potentially bound is not as high, at roughly 130 mg per posologic unit, namely lanthanum carbonate or oxy-hydroxy-sucroferric [63,64]. Hence, it is reasonable that this therapy will be more effective when coupled with controlled phosphate intake. Since dietary protein content is directly associated with that of phosphorus, this means that low-protein regimens are also characterized by a reduced phosphorus content [65], and vice versa.

The benefits coming from low-protein or very-low-protein diets in patients that require phosphate binder therapy is easily understandable and well known in terms of synergic effects and reduction in phosphate binder dosage [66]. Therefore, lower costs and lower rates of gastrointestinal intolerance are expected. Finally, since adherence to therapy is often a matter of the number of pills to be assumed, more compliance to prescriptions is possible. In conclusion, the use of phosphate binders should follow the strategy “less is better”, and the use of a low-protein diet or a very-low-protein diet allows that. Moreover, the benefits of dietary protein restriction can be noticed not only in the control of the laboratory findings, but also in the improvement of osteodystrophy: in particular, the very-low-protein diet supplemented with ketoanalogues is safe if combined with appropriate calcium supplementation, and in combination with vitamin D, which improves both osteitis fibrosa and osteomalacia [67].

Isakova et al. investigated the synergic effect of phosphate binding and phosphate restriction on FGF23 serum levels in a single-blinded, placebo-controlled, 3-month study including CKD patients at stage 3 and 4 and with normal phosphate serum levels. They reported that when an ad libitum diet was compared to a 900 mg phosphate diet, no significant reduction in FGF23 levels occurred; the same was observed when lanthanum carbonate was compared with a placebo. When lanthanum carbonate was associated with a reduction in dietary phosphate, a 35% reduction in FGF23 serum levels was observed by the end of the study period. The authors concluded that only the combination of lanthanum carbonate with a reduction in phosphate intake could decrease FGF23 circulating levels [68].

In another short-term randomized controlled trial, Sigrist et al. compared the effects of a high-phosphate diet (2000 mg/day), low-phosphate diet (750 mg/day) and low-phosphate diet plus an intestinal phosphate binder, namely aluminum hydroxide, at a dosage of 500 mg thrice daily; the two diets supplied comparable macronutrient amounts. Both in CKD and in healthy subjects, serum and urinary phosphate, calcitriol and FGF23 serum levels were measured. Favorable changes in calcitriol and FGF23 serum levels occurred in CKD patients on the low-phosphate diet, as expected. The addition of an intestinal phosphate binder, namely aluminum hydroxide, exerted a synergic action on these two parameters [69].

## 6. Conclusive Considerations

The care management of CKD patients is complex, and clinical studies have often explored the benefits of a single drug or a specific dietetic approach to avoid possible confounding factors. However, daily clinical practice involves comorbid conditions that undergo different pharmacological and dietary treatment. Therefore, it is interesting to consider the potential synergisms and cooperation between diet and drugs.

The transposition “from the bench to bedside” is obviously complex and must be adapted to specific medical and social scenarios. In fact, the results of the trials also depend on the high adherence of the patients and the special attention of the researchers, conditions that are difficult to reproduce in real-world practice. However, nephrologists must know and use all of the arrows in their quiver to optimize and personalize therapy in order to reduce the progression of CKD and dialysis requirement. The use of a dietary approach with RAAS blockers and SGLT2 inhibitors with their synergic effects is the best example of an integrated approach.

The synergism between renal diets and medications in CKD patients may be summarized by the following key points. A high sodium intake blunts the efficacy of RAAS inhibitors on proteinuria, blood pressure and progressive kidney damage.

Sodium restriction improves the anti-proteinuric and anti-hypertensive effect of RAAS inhibitors and therefore further reduces the risk of progressive kidney damage. Sodium restriction increases the anti-proteinuric and anti-hypertensive action of ACE inhibitors much more than the association with ARB (RAAS double block). A protein-restricted diet has an anti-proteinuric effect additive to ACE inhibitors or ARBs. The reno-protective effects of SGLT2 inhibitors are independent of the level of protein intake, so it is speculated that an additive effect might exist when SGLT2 inhibitors are added on top of a protein-restricted diet, or vice versa. The restriction of phosphorus intake in association with intestinal phosphate binders causes a reduction in FGF23 serum levels in CKD patients. The restriction of protein intake ameliorates the bone marrow’s response to epoetin administration.

In conclusion, nutritional and pharmacological therapies represent the basis for the non-dialysis management of CKD patients. Both kinds of treatments have specific and unchangeable features, and, in certain cases, they also have a synergic action (Figure 1). These results will provide an optimization of therapies which is more effective and less costly, and involves a lower risk of unwanted side effects. Nutrition and medications should not be seen as competitive therapies, but rather as a cooperation strategy to lower the risk of progressive kidney and cardiovascular damage, and to delay dialysis initiation.

As occurs for all of the effective therapies, the risk of unwanted side effects is present, and special concerns may arise. For instance, sodium restriction coupled with RAAS inhibition increases the risk of hyponatremia, while a protein-restricted diet coupled with SGLT2 inhibitors caused a risk for a loss of muscle mass. Periodic monitoring of sodium serum levels as well as of muscle mass and body composition should be routinely included in the follow-up of CKD patients. The above-mentioned conditions represent one more reason to implement a close follow-up.

**Figure 1 nutrients-15-02715-f001:**
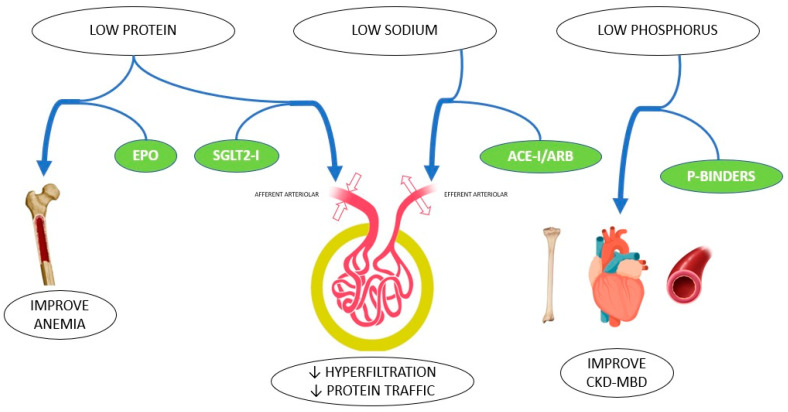
Synergy of nutritional intervention and pharmacological treatment in CKD patients.

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
