# Peer review of "Nutritional Treatment as a Synergic Intervention to Pharmacological Therapy in CKD Patients"

_nutrients, 2023, doi:10.3390/nu15122715_

Round 1
Reviewer 1 Report
Overall, this is a very thoughtful review. However, it would benefit from a more detailed and critical assessment of the impact of low protein diet especially in humans.
Lines 36-41: It is noted here that protein restriction and other dietary modification can “correct uremia-related symptoms…delay dialysis. However, it should be emphasized that protein restriction, at least as far as was achieved in human subjects did not delay “progression” of disease as much as was anticipated from pre-clinical studies and clinical reports in small populations (Line 46).
Lines 51-57: It should be noted that in Ref. 8, plasma protein concentration was not different between the normal and low protein diets and that the changes were attributed by the authors on the higher glomerular pressures. It is basically a chicken and egg phenomenon. If the tubular protein promotes inflammatory and profibrotic upregulation but the increased tubular proteins are due to increased glomerular pressures, which was the initiating factor.
Line 126: There is no reference 120. It seems “20” was intended. Please address the issue that studies with dual ACEi/ARB blockade have since been found to be highly questionable aside from this study and is no longer recommended.
Lines 136-140: Please cite the original articles re: eicosanoids and nitric oxide rather than a review. Citation of a review by a review should be avoided.
Lines 153ff: The authors should address the feasibility to have individuals adhere to a 0.4 g/kg/d protein diet as well as the impact on nutritional status.
Line 173: There is no Ref 230. Should be “23.”
Lines 178-179: The study by Aart-van der Beek et al was interesting in that the median protein intakes were not low: 58.4, 63.6, and 90.0 g/kd/d. In fact, the interesting issue was that the highest (90 g/kg/d) was interpreted by those authors that dietary protein did not impact the effects of the flozins on proteinuria. In fact, the impact on eGFR was no different. Some of the groups were very small. Stating that there was a “trend” may not be strong enough to provide caution.
Line 311-312: Is the antiproteinuric effect of sodium restriction correlated to the greater anti-hypertensive effect on the low sodium diet? This should be addressed.
Minor issues:
1. No need to capitalized Renin Angiotensin Aldosterone System.
Author Response
Reviewer no. 1
Overall, this is a very thoughtful review. However, it would benefit from a more detailed and critical assessment of the impact of low protein diet especially in humans.
Lines 36-41: It is noted here that protein restriction and other dietary modification can “correct uremia-related symptoms…delay dialysis. However, it should be emphasized that protein restriction, at least as far as was achieved in human subjects did not delay “progression” of disease as much as was anticipated from pre-clinical studies and clinical reports in small populations (Line 46).
- A. thank you. We added the following sentence: “Instead, protein restriction seemed to have a statistically significant but small and clinically negligible effect on slowing the rate of decline of glomerular filtration rate (GFR) “.
Lines 51-57: It should be noted that in Ref. 8, plasma protein concentration was not different between the normal and low protein diets and that the changes were attributed by the authors on the higher glomerular pressures. It is basically a chicken and egg phenomenon. If the tubular protein promotes inflammatory and profibrotic upregulation but the increased tubular proteins are due to increased glomerular pressures, which was the initiating factor.
- I fully agree that no changes in plasma protein occur. The increase of protein within the tubular lumen is due to higher glomerular pressure, and the consequent tubular reabsorption elicits pro-inflammatory reaction.
Line 126: There is no reference 120. It seems “20” was intended. Please address the issue that studies with dual ACEi/ARB blockade have since been found to be highly questionable aside from this study and is no longer recommended.
- Sorry for the typo and thank you for raising this point. It is noteworthy that dual ACEi/ARB blockade is highly questionable and no longer recommended in CKD patients especially when diabetics, mainly for increased risk of hyperkalemia.
Lines 136-140: Please cite the original articles re: eicosanoids and nitric oxide rather than a review. Citation of a review by a review should be avoided.
- You are right. The following citations have been added;
Don, B.R,; Blake, S.; Hutchison, F.N.; Kaysen, G.A.; Schambelan, M. Dietary protein intake modulates glomerular eicosanoid production in the rat. Am J Physiol. 1989,256(4 Pt 2):F711–8
King, A.J.; Troy, J.L.; Anderson, S.; Neuringer, J.R.; Gunning, M.; Brenner, B.M. Nitric oxide: a potential mediator of amino acid-induced renal hyperemia and hyperfiltration. J Am Soc Nephrol 1991;1(12): 1271–7
Lines 153ff: The authors should address the feasibility to have individuals adhere to a 0.4 g/kg/d protein diet as well as the impact on nutritional status.
A . It is conceivable that very low protein diets are difficult to adhere, this is the reason why they are proposed to motivated and selected patients. When diet management is correct no signs of malnutrition occur.
Line 173: There is no Ref 230. Should be “23.”
- We apologize for the typo
Lines 178-179: The study by Aart-van der Beek et al was interesting in that the median protein intakes were not low: 58.4, 63.6, and 90.0 g/kd/d. In fact, the interesting issue was that the highest (90 g/kg/d) was interpreted by those authors that dietary protein did not impact the effects of the flozins on proteinuria. In fact, the impact on eGFR was no different. Some of the groups were very small. Stating that there was a “trend” may not be strong enough to provide caution.
- I agree with your comments, and the sentences have been edited accordingly.
Line 311-312: Is the antiproteinuric effect of sodium restriction correlated to the greater anti-hypertensive effect on the low sodium diet? This should be addressed.
A. Low sodium diet exert anti-proteinuric and anti-hypertensive effects, but lowering proteinuria by sodium restriction was independent of the changes of arterial blood pressure [16 ]
Minor issues:
No need to capitalized Renin Angiotensin Aldosterone System.
A. Ok, it has been corrected
Reviewer 2 Report
The authors' review of the effects of combination drug therapy with nutrition treatment, specifically focusing on salt and RAAS inhibition, SGLT2i, ESA, and phosphorus, is an important and intriguing aspect of medical research. However, it is crucial to note that the manuscript solely discusses the benefits of this combination without addressing the potential risks of adverse effects associated with such a strategy. Hence, I highly recommend that the manuscript be updated to include a comprehensive analysis of both the benefits and risks involved.
Major points:
One notable concern is the possibility of hyponatremia arising from RAAS inhibition and excessive salt restriction, particularly in patients with advanced stage chronic kidney disease (CKD). Hyponatremia can lead to various significant problems, such as nausea, headache, confusion, fatigue, and even bone-related complications in the long term. It is essential for the authors to reference and discuss these aspects in their work.
While SGLT2i shows promise in managing CKD progression, it is important to acknowledge that it can also contribute to muscle loss and promote a catabolic state in terms of protein metabolism. Consequently, recommending the use of SGLT2i alongside protein restriction, particularly in elderly patients with CKD, carries the risk of frailty. The authors should address this concern in their manuscript.
Moreover, the restriction of phosphorus intake and the utilization of phosphorus binders can potentially lead to hypophosphatemia. This condition is associated with muscle weakness, and chronic hypophosphatemia can also contribute to bone-related issues.
Minor points:
The authors should ensure the accurate correction of reference numbers throughout the manuscript. Several mistakes, such as referencing "120" instead of "12," have been identified and need to be rectified.
Furthermore, the conclusion section of the manuscript presents some issues in terms of formatting and coherence. The inconsistent font and abrupt line changes create an unnatural and disjointed appearance.
In conclusion, while the authors' focus on the combination of drug therapy with nutrition treatment is commendable, it is imperative that they expand their discussion to encompass the potential adverse effects associated with this approach. By addressing these concerns, the manuscript will provide a more comprehensive and balanced perspective on the subject matter.
Author Response
The authors' review of the effects of combination drug therapy with nutrition treatment, specifically focusing on salt and RAAS inhibition, SGLT2i, ESA, and phosphorus, is an important and intriguing aspect of medical research. However, it is crucial to note that the manuscript solely discusses the benefits of this combination without addressing the potential risks of adverse effects associated with such a strategy. Hence, I highly recommend that the manuscript be updated to include a comprehensive analysis of both the benefits and risks involved.
- this is an interesting point. I will discuss this point.
Major points:
One notable concern is the possibility of hyponatremia arising from RAAS inhibition and excessive salt restriction, particularly in patients with advanced stage chronic kidney disease (CKD). Hyponatremia can lead to various significant problems, such as nausea, headache, confusion, fatigue, and even bone-related complications in the long term. It is essential for the authors to reference and discuss these aspects in their work.
A. Thank you for this comment. Lowering sodium serum levels may occur in patients taking ACEI or ARB. As well, low salt intake can lead to hyponatremia but when associated with high water intake, leading to an impairment between water and sodium balance and hence of body water homeostasis. The risk of hyponatriemia is more consistent when low sodium diet is associated with high dose of diuretics. Monitoring of sodium serum level is recommended in the patients with ACE inhibitor or ARB administration and on low sodium diet. We did not find a special warning in the literature about the association between RAAS inhibitor and sodium restriction which, o the contrary, is strongly recommended.
The following citations have been added in the reference list
Liamis G, Megapanou E, Elisaf M, Milionis H. Hyponatremia-Inducing Drugs. Front Horm Res. 2019;52:167-177. doi: 10.1159/000493246.
Hooper L, Bartlett C, Davey Smith G, Ebrahim S. The long term effects of advice to cut down on salt in food on deaths, cardiovascular disease and blood pressure in adults. Cochrane Database Syst Rev 2004; (4) Art No CD003177
While SGLT2i shows promise in managing CKD progression, it is important to acknowledge that it can also contribute to muscle loss and promote a catabolic state in terms of protein metabolism. Consequently, recommending the use of SGLT2i alongside protein restriction, particularly in elderly patients with CKD, carries the risk of frailty. The authors should address this concern in their manuscript.
- Thank you for this comment. The effects of SGLT2 inhibition on muscle mass are not yet definite; decrease or no change of skeletal muscle mass have been reported. Negative energy balance increases circulating amino acids concentrations and stimulates amino acid oxidation . These findings may be expression of protein catabolism similarly to the increased fat oxidation. Dapagliflozin treatment causes a reduction in body weight mainly by reducing fat mass, but some changes are reported also on amino acid metabolism [Horibe].
The study by Horibe et al. reported that the addition of dapagliflozin significantly reduced body and fat mass in patients with type 2 diabetes not receiving insulin. Instead, skeletal muscle mass showed no significant reduction after treatment with dapagliflozin for 24 weeks. Finally, changes in the plasma amino acid profile were detected after 24 week treatment with dapagliflozin.
A reduction in skeletal muscle mass by SGLT2 inhibitors might have a negative effect in older or lean patients with type 2 diabetes.
The following citation has been added in the reference list.
Horibe K et al. Metabolic changes induced by dapagliflozin, an SGLT2 inhibitor, in Japanese patients with type 2 diabetes treated by oral anti-diabetic agents: A randomized, clinical trial. Diabetes Res Clin Pract 2022 Apr;186:109781
Moreover, the restriction of phosphorus intake and the utilization of phosphorus binders can potentially lead to hypophosphatemia. This condition is associated with muscle weakness, and chronic hypophosphatemia can also contribute to bone-related issues.
A…in my opinion this is an unlikely concern. The use of phosphate binders in CKD is indicated only in case of hyperphosphatemia. Hypophosphatemia never occurs in CKD apart from cases of severe malnutrition or proximal tubule diseases: in these cases there is no suggestions for both phosphate binders and/or low protein low-phosphorus diet. In any case, monitoring of phosphate and CKD-MBD parameters is a measure of good clinical practice in CKD clinic.
Minor points:
The authors should ensure the accurate correction of reference numbers throughout the manuscript. Several mistakes, such as referencing "120" instead of "12," have been identified and need to be rectified.
- We apologize for the typos. They have been corrected.
Furthermore, the conclusion section of the manuscript presents some issues in terms of formatting and coherence. The inconsistent font and abrupt line changes create an unnatural and disjointed appearance.
- We aimed to underline the key points of the paper. However, accordingly to your comments, the format of the conclusion section has been uniformed with all the text
In conclusion, while the authors' focus on the combination of drug therapy with nutrition treatment is commendable, it is imperative that they expand their discussion to encompass the potential adverse effects associated with this approach. By addressing these concerns, the manuscript will provide a more comprehensive and balanced perspective on the subject matter.
- A. Thank you for your criticisms. The potential adverse effects are those commonly described for the cited medications and also for the dietary interventions. There are no proven serious adverse effect raising from the associations analyzed in the manuscript. However, some hypotheses have been reported and discussed
Round 2
Reviewer 2 Report
The authors addressed my concerns.
So I have no further comments for this review article.